# Deep Learning Network of *Amomum villosum* Quality Classification and Origin Identification Based on X-ray Technology

**DOI:** 10.3390/foods12091775

**Published:** 2023-04-25

**Authors:** Zhouyou Wu, Qilong Xue, Peiqi Miao, Chenfei Li, Xinlong Liu, Yukang Cheng, Kunhong Miao, Yang Yu, Zheng Li

**Affiliations:** 1Xin-Huangpu Joint Innovation Institute of Chinese Medicine, Guangzhou 510715, China; wzy210426@163.com (Z.W.); xql_975@163.com (Q.X.); miaopeiqi@163.com (P.M.); lcf_0327@163.com (C.L.); 19999442433@163.com (X.L.); cyk_gift@163.com (Y.C.); lizheng@tjutcm.edu.cn (Z.L.); 2State Key Laboratory of Component Traditional Chinese Medicine, Tianjin 301617, China; 3College of Pharmaceutical Engineering of Traditional Chinese Medicine, Tianjin University of Traditional Chinese Medicine, Tianjin 301617, China; 4Tianjin Modern Innovative TCM Technology Co., Ltd., Tianjin 300380, China; 5Haihe Laboratory of Modern Chinese Medicine, Tianjin 301617, China

**Keywords:** non-destructive detection, X-ray imaging, convolutional, *Amomum villosum* fruit

## Abstract

A machine vision system based on a convolutional neural network (CNN) was proposed to sort *Amomum villosum* using X-ray non-destructive testing technology in this study. The *Amomum villosum* fruit network (AFNet) algorithm was developed to identify the internal structure for quality classification and origin identification in this manuscript. This network model is composed of experimental features of *Amomum villosum*. In this study, we adopted a binary classification method twice consecutive to identify the origin and quality of *Amomum villosum*. The results show that the accuracy, precision, and specificity of the AFNet for quality classification were 96.33%, 96.27%, and 100.0%, respectively, achieving higher accuracy than traditional CNN under the condition of faster operation speed. In addition, the model can also achieve an accuracy of 90.60% for the identification of places of origin. The accuracy of multi-category classification performed later with the consistent network structure is lower than that of the cascaded CNNs solution. With this intelligent feature recognition model, the internal structure information of *Amomum villosum* can be determined based on X-ray technology. Its application will play a positive role to improve industrial production efficiency.

## 1. Introduction

*Amomum villosum* has been traditionally used as a food flavoring [1,2]. The plant is distributed from Sri Lanka to the Himalayas, China, southeast Asia, Malaysia, and northern Australia [3]. Pharmacological studies showed that *Amomum villosum* has great anti-diarrheal, anti-ulcer, anti-inflammatory, and antibacterial effects [4]. Furthermore, *Amomum villosum* has also been commonly used in Chinese cooking as a spice to mask the fishy smell of meat [1]. However, the quality of *Amomum villosum* in the market needs to be better controlled to ensure its safe use [5]. *Amomum villosum* may become moldy, deteriorate, or burst during harvesting, transportation, and processing. This results in a mixed quality of *Amomum villosum* sold in the market [1]. To control its quality, Fourier transform near-infrared spectroscopy and gas chromatography was used to determine the sample of *Amomum villosum* [6].

However, traditional analysis methods cannot achieve rapid and non-destructive detection of *Amomum villosum*. In recent years, the machine vision approach has been widely employed in the quality inspection of vegetables and other foods. For example, a tomato grading machine vision system performed calyx and stalk scar detection at an average accuracy of 0.9515 for both defective and healthy tomatoes [7]. An automatic carrot grading system was developed to inspect the surface quality of carrots based on computer vision and deep learning [8]. An automatic sorting system was developed to detect the quality of fresh white button mushrooms based on image processing [9]. The studies above illustrate that machine visioning can be employed to detect defective foods intelligently. However, most of the defects of *Amomum villosum* are hard to identify from the outside. Therefore, a non-destructive detecting technology is urgently needed to identify internal defects to ensure quality. Convolutional neural networks have rapidly advanced in various fields within the past few years [10]. X-ray technology has been successfully combined with CNN for fish bone detection [11,12], radiological image analysis [10], and chest X-ray examination [13,14]. Low-power X-ray detection technology could be used to identify the internal characteristics of fruits within safety limits. Images of internal defects of food can be acquired using X-ray technology, and the food can then be classified as normal or defective based on the images. For example, X-ray technology combined with CNN can be used to locate seed spoilage in mango fruit [15]. In addition, X-ray imaging technology is also employed to inspect onions [16], deboned poultry [17], etc. Most studies identify the origin based on the analysis of the components, such as tea [18], orange [19], and P. notoginseng main roots [20]. In addition, quality defects were distinguished in most studies, but origin identification has often been overlooked, such as in the classification of mangoes [21], eggs [22], apples [23], etc. Generally, the components of plants will vary due to temperature, precipitation, climate, soil characteristics, and other conditions when they are from different places of origin. Such plants include Gentiana rigescens Franch [24], sesame seeds [25], and Lonicerae Japonicae Flos [26]. Therefore, identifying the origin of *Amomum villosum* based on X-ray technology is beneficial to maintain stable product quality.

In this study, we explored the possibility of identifying the origin of *Amomum villosum* directly from the images. A non-destructive inspection method was developed to detect the defective *Amomum villosum* fruits, which combines deep learning and X-ray imaging technology. A convolution neural network was developed to improve the accuracy of evaluating the quality of *Amomum villosum*. A new algorithm was developed to identify the origin of *Amomum villosum*, which achieved higher accuracy than the traditional convolutional neural network. 

## 2. Materials and Methods

### 2.1. Samples

The *Amomum villosum* fruit was provided by Tianjin Shangyaotang Pharmaceutical Co., Ltd. (Tianjin, China). The samples were collected from two places of origin, Yunnan and Guangdong. All of them were divided into normal and defective categories according to their X-ray images through manual inspection.

### 2.2. X-ray Detection System

In this study, the experimental system, which has been used to capture the X-ray images of sterculia seeds, was used to obtain X-ray images of *Amomum villosum* to perform future research on origin identification [27]. In the online X-ray imaging system, a linear array X-ray detector was used to capture the gray lines generated by the X-rays passing through the fruits of *Amomum villosum*.

The image acquisition device consists of the following parts: servo driver, servo motor, X-ray source, linear array detector. The model of the servo driver is MADLT15SF (Panasonic Industry Co., Ltd., Tokyo, Japan) and the frequency response is set at 3.2 kHz. Meanwhile, the supply voltage is single/3-phase 200 V. The I/F classification of type is analog/pulse when the protocol is Modbus and the interface is RS485/RS232. The model of the servo motor is MHMF022L1U2M (Panasonic Industry Co., Ltd.). Its rated output is 200.0 W and the rated current is 1.4 A (rm s); moreover, the rated speed is 3000.0 r/min and the rated torque is 0.64 N m.

The X-ray source adopts the mini-focus X-ray system V.J IXS0808. Its output current is 0.2–0.7 mA and the output voltage is 20–80 KV. In addition, the maximum continuous output power is 50 W. 

The linear array detector employed the data acquisition system XNDT and the technology is XNDT-04. It has a scanning range of 400–600 mm and the pixel size is between 50 μm and 0.8 mm. Furthermore, the SNR system is 25,000:1 and the electronic SNR is 50,000:1.

The acquired digital image is transmitted to the workstation through the network cable. The control interface program and hardware driver program were designed with C#.NET 4.0 (visual studio 2017).

### 2.3. X-ray Image Acquisition and Pre-Processing

The difference of gray value in the X-ray image can be explained by the principle of X-ray attenuation, as shown in Equation (1).
(1)$$\frac{I}{I_{0}}=EXP(-\mu_{m}\rho L)$$

In Equation (1), *I*, *I*_0_, and *µ_m_* represent the transmitted effective X-ray intensity, incident effective X-ray intensity, and the linear attenuation coefficient (cm^−1^), respectively. ρ is the material density and L is the material thickness. X-ray attenuation occurs through the fruit, and the degree of attenuation depends on the thickness and density of the material.

*Amomum villosum* fruit was placed on the conveyor belt and its X-ray image was photographed and saved on the computer. Next, 1600 X-ray images were manually divided into normal and defect groups for their quality. At the same time, they were also divided into the Yunnan and Guangdong groups for the study of their place of origin. All of the images are shown in Figure 1 for the various groups. 

Figure 1a shows the normal sample produced in Yunnan. Meanwhile, Figure 1b shows the X-ray image of the normal sample produced in Guangdong. The shapes of the normal samples from various origins are different. For Figure 1a, the inner seed kernel is larger than that in Figure 1b. On the contrary, the X-ray image of the defective fruit samples produced in the same location has a blurred outer contour and small or no inner nuts in Figure 1c. 

As shown in Figure 1, it can be found that there is an apparent structural difference between the normal and defect groups by the visual inspection of the X-ray images. For the normal groups, it is possible to distinguish various places of origin based on their appearance.

Based on the differences between the normal and defective images mentioned above, as well as the differences between the images of fruits from different regions, the dataset was divided into quality classification and region classification. It is worth mentioning that in order to explore the possibility of distinguishing fruit images of different origins and qualities at once, considering that there is no considerable difference in the images of defective fruits from different regions, the previous data was further divided into three classification datasets.

After labeling, the images of different datasets were divided into training sets, validation sets, and testing sets according to Table 1. From Table 1, it can be seen that the quality classification dataset has a total of 1600 images, which are divided into normal and defective according to the quality, with 1430 and 170 images, respectively. In addition, it is divided into training, validation, and testing sets according to the ratio of 3:1:1. The origin identification dataset contains 455 and 594 images of fruit from Yunnan and Guangdong, respectively, and only contains normal images from different origins after removing the defective images and atypical images. Finally, is the images were divided into training, validation, and test set according to the ratio of 7:2:2. The three-category classification dataset contains 452, 587, and 101 normal Yunnan, normal Guangdong, and defective images, respectively. It is divided into training, validation, and test sets according to the ratio of 4:1:1.

Before the quality classification, the fruit images were randomly converted and changed to increase the varieties of the dataset. The X-ray images were randomly rotated by 30° or moved horizontally and vertically by 10%. Then, they were flipped horizontally. Due to the original image size being around 400 pixels × 400 pixels and variable, all the images were uniformly adjusted to 50 pixels × 50 pixels to adapt to the later network structure. Finally, all the images were adjusted to 50 pixels × 50 pixels to adapt to the later network structure. After quality classification, the images of the normal fruits that were identified by the CNN model were preprocessed and used for origin identification. Filler white edges were added around the original images to make each image become 400 pixels × 400 pixels to avoid deforming the images in the deformation steps before the study of origin identification. The fruit from the Yunnan group was labeled as 1, and the fruit from the Guangdong group was labeled as 0.

### 2.4. The Analysis Process and Network Structure

#### 2.4.1. The Analysis Process

In this paper, the network structure used to detect defective *Amomum villosum* is the same as that used to detect the place of origin of *Amomum villosum*. The methodology diagram is shown in Figure 2. Firstly, the X-ray images of *Amomum villosum* from various places of origin were input into the network. Next, the detected defective fruit was removed from the input. The remaining normal fruits were fed into the network model to identify the place of origin. The solid arrow represents the connection of the network layer. The dotted arrow indicates that the two networks have the same structure. The proposed AFNet architecture for non-destructive testing was implemented in Python 3.7 using the Tensorflow backend. The training process was implemented on a Windows 10 system with a 3080Ti GPU.

#### 2.4.2. Structure of the Proposed *Amomum villosum* Fruit Network 

As shown in Table 2, a convolutional neural network including 3 convolution layers and 3 full connection layers was used in this study. The convolutional layers consist of 4, 8, and 16 filters with a filter size of 3 × 3. A 2 × 2 max-pooling function was used to modify the output results of the convolutional layer. In the AFNet, the L2 regularization method and dropout layer was used to prevent overfitting. The set strategy of the dropout rate for each layer is shown in Table 2. In this work, the rectified linear unit (ReLU) was used as an activation function overall in the network. In the output layer, a Softmax activation was employed to solve the binary classification problem. Sparse categorical cross-entropy loss function was used to train the network. 

### 2.5. Quality Classification of Amomum villosum Fruit 

#### Training Process of Quality Classification 

In this study, the X-ray images of *Amomum villosum* fruit from two places of origin were mixed and then divided into two groups, namely, normal and defective by quality. In the normal group, the samples must have full contents. On the contrary, others can be classified as defective. Then, the neural network was trained to divide the input fruit pictures into normal and defective ones and output them as results.

### 2.6. Origin Identification 

#### Training Process of Origin Identification

After the study of quality classification, the normal fruit images were sent to the network again for training to distinguish the place of origin of *Amomum villosum*. The preprocessed images were mixed into training set and validation set, and the validation set was randomly shuffled to ensure that the neural network encountered images of various labels. 

### 2.7. The Multi-Category Classification of Amomum villosum Fruits

#### 2.7.1. Training Process of Multi-Category Classification

The obtained X-ray images were placed directly into the AFNet model for multi-category classification in the following research, as shown in Figure 3. The X-ray images of *Amomum villosum* from the two places of origin with a mixture of qualities were divided into three types. The type of non-defective sample from Yunnan, the type of non-defective sample from Guangdong, and the defective sample were labeled as 0, 1, and 2, respectively. After that, the images of *Amomum villosum* fruit were sent to the same neural network as the binary classification for prediction.

#### 2.7.2. Parameters Optimization 

All the X-ray images of *Amomum villosum* were input into the convolutional neural network model. In addition, the network structure was modified to achieve the optimal classification performance. At the same time, the hyperparameters of the network model were adjusted to achieve the highest accuracy, as well as the binary classification. The optimal prediction performance was obtained while the model was kept consistent with the original network structure. The batch size among the four levels was optimized as a value between 64 and 8. Furthermore, the learning rate among the five levels was adjusted between 5.0 × 10^−4^ and 1.0 × 10^−5^.

### 2.8. Model Comparison 

Traditional CNN models VGG16, ResNet18, and Inception were used to compare the classification performance with the AFNet model. The accuracy, precision, and specificity were used to measure their performance. The same parameters and data were used for training and testing. In this work of quality classification, the batch size was set as 32, the learning rate was set as 8.0 × 10^−5^, and the epoch was set as 300.

### 2.9. Evaluation Standards 

The binary classification performance of AFNet is measured by three criteria: accuracy, specificity, and precision. These parameters were calculated by the following Equations (2)–(4) [28]:(2)$$Accuracy=\frac{TP+TN}{TP+TN+FP+FN}$$
(3)$$Precision=\frac{TP}{TP+FP}$$
(4)$$Specificity=\frac{TP}{TP+FN}$$

TP, FN, FP, and TN represent the number of true positives, false negatives, false positives, and true negatives, respectively.

The multi-class classification performance of AFNet is measured by three standards: average accuracy, precision, and recall. These parameters are calculated using the following Equations (5)–(7) [29]:(5)$$\mathrm{Average}\;\mathrm{accuracy}=\frac{\sum_{i=1}^{l}\frac{tp_{i}+tn_{i}}{tp_{i}+fn_{i}+fp_{i}+tn_{i}}}{l}$$
(6)$${\mathrm{Precision}}_{M}=\frac{\sum_{i=1}^{l}\frac{tp_{i}}{tp_{i}+fp_{i}}}{l}$$
(7)$${\mathrm{Recall}}_{M}=\frac{\sum_{i=1}^{l}\frac{tp_{i}}{tp_{i}+fn_{i}}}{l}$$

*Tp_i_, fp_i_, fn_i_*, and *tn_i_* are true positive, false positive, false negative, and true negative counts for Ci, respectively. M indices represent macro-averaging.

## 3. Results

### 3.1. Performance of AFNet in Detecting Defective Fruits

#### Confusion Matrix of the Validation Dataset

The robustness of the AFNet model was tested using a validation set containing 386 X-ray images of *Amomum villosum* from different place of origin. As shown in Figure 4, the accuracy, precision, and specificity of AFNet can be calculated as 96.33%, 96.27%, and 100.0%, respectively, according to the confusion matrix.

### 3.2. The Performance of AFNet in Distinguishing Places of Origin

#### 3.2.1. Accuracy and Loss Curve

As shown in Figure 5, the accuracy rate of the training set shows a gradual upward trend with the increase of iterations. The accuracy of the validation set fluctuated violently in the first 200 epochs, and gradually stabilized after 200 epochs and fitted the accuracy of the training datasets. The accuracy reached the highest value at the 300th epoch. In the first 50 epochs, the loss function curve of the training dataset decreased rapidly. The decline rate of the loss function curve of the training dataset before 300 epochs slowed down. The loss function curve of the validation dataset decreased rapidly in the first 100 epochs and tended to be gentle before 300 epochs. Overall, the curves of the validation dataset and training dataset are relatively close, indicating that there is no or slight overfitting.

#### 3.2.2. Confusion Matrix of the Validation Dataset

An independent dataset containing 149 X-ray images of *Amomum villosum* was used to validate the robustness of the proposed model. As shown in Figure 6, the confusion matrix was drawn on the basis of the obtained results. According to the confusion matrix, the accuracy, precision, and specificity of AFNet in identifying the place of origin can be calculated as 90.60%, 91.11%, and 80.39%, respectively.

### 3.3. Performance of AFNet in Multi-Category Classification

#### 3.3.1. Parameter Optimization 

Figure 7 shows that the relationship of accuracy changed with the learning rate. When the learning rate was set to 8 × 10^−5^, the accuracy rate reached the highest. Then, the batch size was set to 32, followed by a batch size set to 64.

Figure 8 shows the condition of the batch size that was set to 32. As the learning rate gradually decreased, the accuracy reached the peak with the condition that the learning rate was set to 8 × 10^−5^. Therefore, the parameter configuration achieved optimal network performance when the batch size and the learning rate were set to 32 and 8 × 10^−5^, respectively.

#### 3.3.2. Accuracy and Loss Curve 

As can be seen from Figure 9, the overall accuracy curves dramatically increased and remained stable at around 0.9 after about 300 iterations. The loss curves of the training and validation datasets declined dramatically after the first 50 iterations. The gap between the validation and training loss curves tended to stabilize after epoch 50.

#### 3.3.3. Confusion Matrix of the Validation Dataset

An independent dataset containing 209 X-ray images of *Amomum villosum* was used to validate the robustness of the proposed model. Figure 10 reports that the confusion matrix was obtained for the multi-category grading model. According to the confusion matrix, the average accuracy, precision, and recall of AFNet in multi-category was 90.08%, 86.48%, and 88.47%, respectively.

### 3.4. Comparison with Traditional CNN Model

A validation dataset containing 386 X-ray images of *Amomum villosum* was used to test the performance of the AFNet model and traditional CNN models. As shown in Figure 4, the confusion matrix was drawn based on the experimental results. As shown in Table 3, AFNet achieved similar or higher accuracy than the traditional CNN model.

## 4. Discussion

In this study, a convolutional neural network model based on X-ray technology was established with the deep learning method. The model performed well for both quality classification and origin identification. In terms of quality classification, AFNet achieved similar accuracy compared with the traditional algorithm methods. Furthermore, it will be suitable for industrial production scenarios that require higher detection speed when its network layers are significantly less than traditional algorithm methods. In terms of origin identification, this model can be used to detect the origins of *Amomum villosum* not limited to Yunnan and Guangdong. It has also achieved the desired classification effect. Moreover, as was mentioned in the introduction, there are differences in the fruit components of various origins, which further affects the quality of downstream products. Therefore, this model can be used to control the fruit quality, preferably through the identification of the origin, which is also the novelty of this approach. However, 10 non-defective samples from Yunnan were wrongly classified. The reason for this might be that the images from another place of origin were substantially identical and only different in contour shape. Therefore, the number of samples in the training dataset can be increased to improve the accuracy of origin identification in the future. Moreover, CT-scanning can be used to create artificial X-ray images and reduce data acquisition [30]. In the actual production process, the disordered background may interfere with the detection of the internal information of the *Amomum villosum*. Therefore, we can use image preprocessing methods, such as image binarization, to extract the regions of interest before detection.

In the muti-category classification experiment added in this paper, the number of original network layers was adjusted accordingly to achieve optimal performance. However, the final result shows that the accuracy of multi-classification is lower than the cascaded CNNs solution.

With improved model accuracy, the detection method can potentially switch from one-by-one-sample mode to multiple-samples-at-a-time mode to improve the efficiency of classification. At the same time, it can also be combined with other detection methods to capture different images, such as the RGB image of the fruit to determine the external rot, mildew, and other information about the fruit. A variety of methods have been adopted to detect food quality in previous studies. In the case of orange juice, double-effect real-time PCR was used to exploit the adulteration of mandarin juice [31]. Sandra et al. used hyperspectral images to detect the internal mechanical damage of persimmon fruit [32]. Furthermore, compared with previous research, the method adopted in this study has dramatically improved the detection speed while also ensuring accuracy, and can more readily adapt to the needs of industrial online detection. In addition, sampling inspection has been adopted to control fruits in some studies. However, the full detection method that can control the quality of the final product more comprehensively was adopted for the fruit in this paper. In addition, this experiment explored how to use a single model to classify *Amomum villosum* images on the basis of multi-category training. It provides a reference for the classification of other fruits in various places of origin. In the future, this algorithm can be combined with new technologies, such as hyperspectral image analysis, to detect the internal components of fruit and be applied to other kernels, such as peach kernels and walnuts. This model can also be applied to the quality inspection of food, wood, and other items with internal defects to meet the needs of online sorting and replace manual quality detection in various industrial scenarios.

## 5. Conclusions

Based on CNN deep learning and X-ray technology, this study developed a new model for rapid nondestructive quality classification and origin identification of *Amomum villosum* fruits. A total of 1600 X-ray images were used to train and test the proposed model. The accuracy of quality classification can reach 96.33%. Meanwhile, the accuracy of origin identification can reach 90.60%. The developed model can potentially be applied in the industrial production process to improve accuracy and efficiency.

## Figures and Tables

**Figure 1 foods-12-01775-f001:**
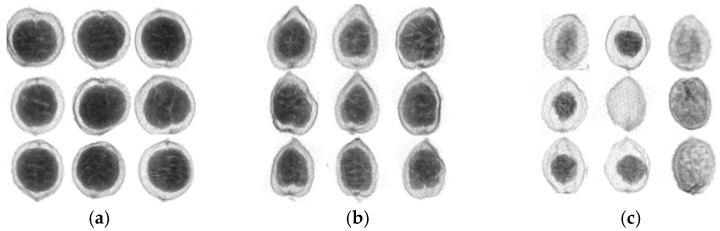
X-ray images of normal and defective samples in Yunnan and Guangdong. (**a**) Normal samples from Yunnan; (**b**) normal samples from Guangdong; (**c**) defective samples.

**Figure 2 foods-12-01775-f002:**
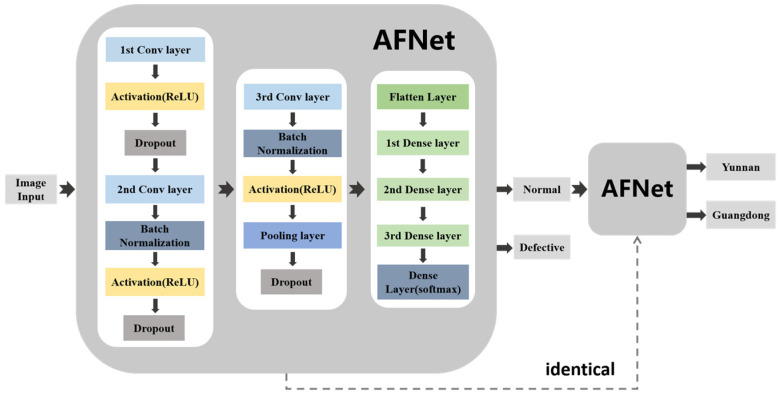
The diagram of the analysis process of the overall architecture of AFNet.

**Figure 3 foods-12-01775-f003:**
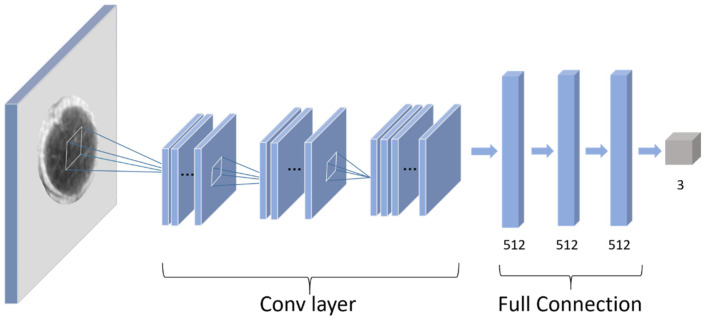
Flowchart of the proposed multi-category classification.

**Figure 4 foods-12-01775-f004:**
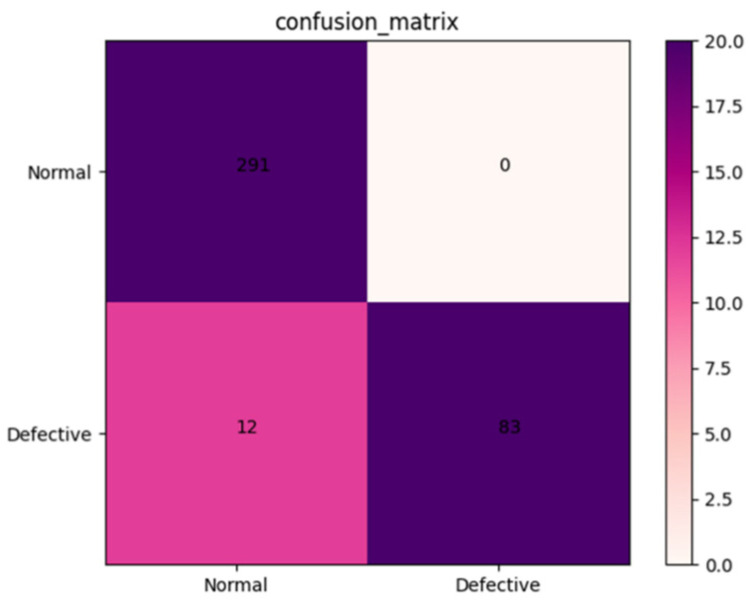
Confusion matrix of the proposed AFNet model when detecting normal and defective samples from various places.

**Figure 5 foods-12-01775-f005:**
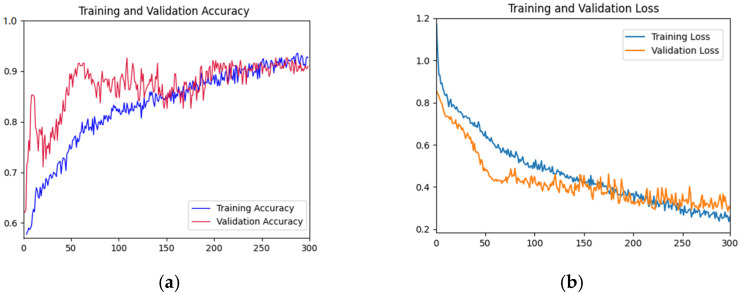
Training and testing curves of AFNet in distinguishing the place of origin of the fruits. (**a**) Accuracy curves; (**b**) Loss curves. In Figure 5, the *x*-axis is the epochs; in (**a**), the *y*-axis is accuracy; and in (**b**), the *y*-axis is the loss value.

**Figure 6 foods-12-01775-f006:**
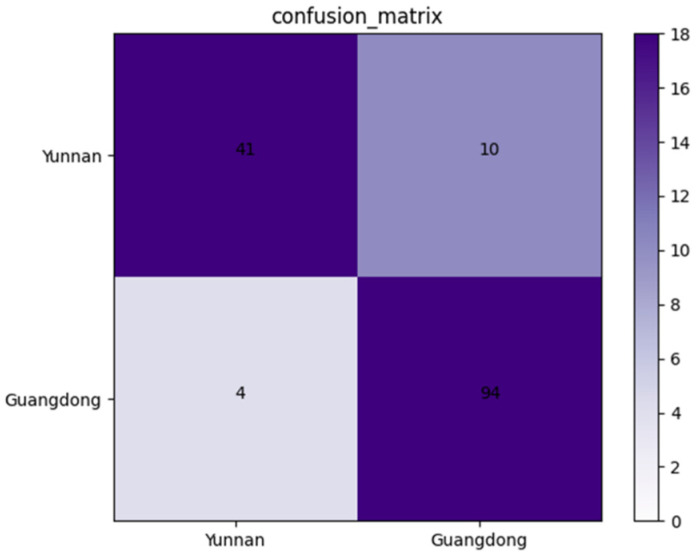
Confusion matrix of the proposed AFNet model when distinguishing various places of origin.

**Figure 7 foods-12-01775-f007:**
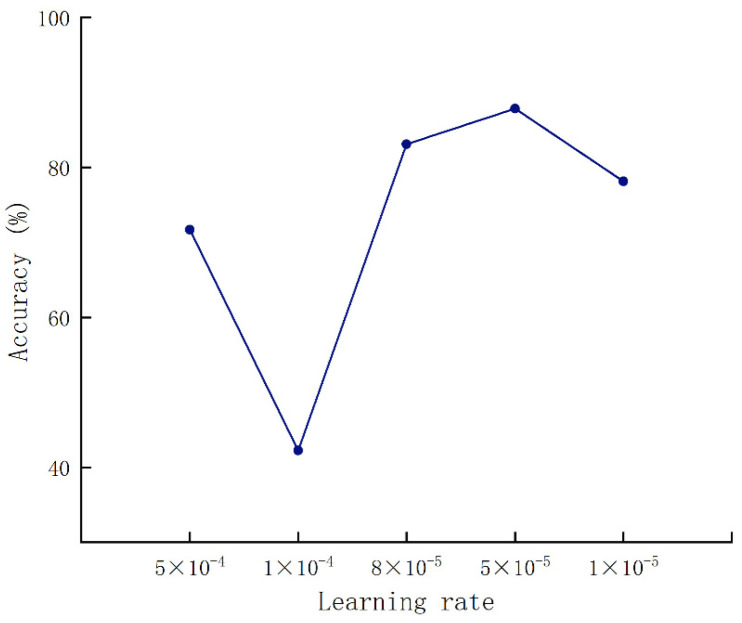
Validation accuracy results under various learning rates.

**Figure 8 foods-12-01775-f008:**
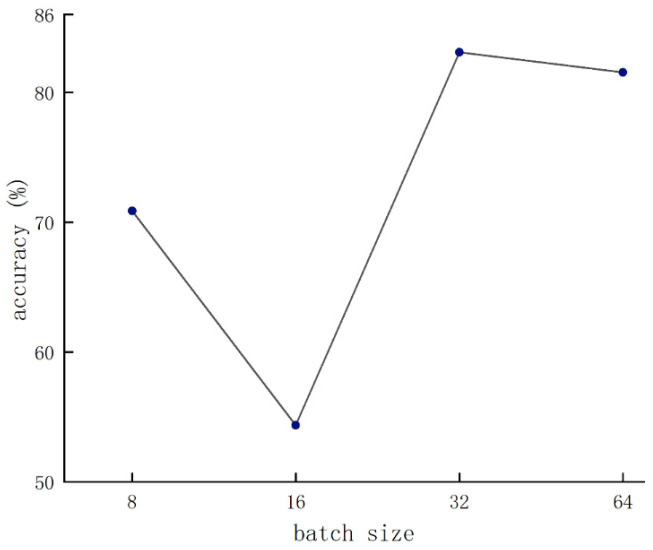
Validation results of various batch sizes.

**Figure 9 foods-12-01775-f009:**
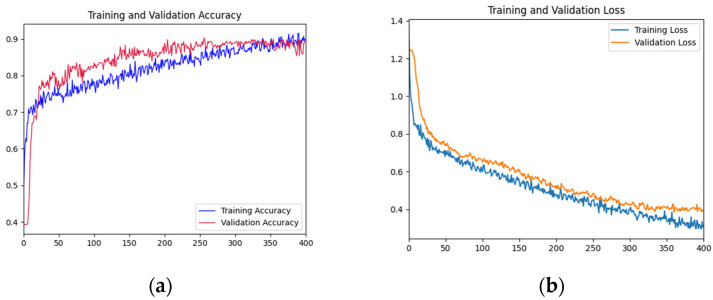
Training and testing curves of AFNet in muti-category. (**a**) Accuracy curves; (**b**) Loss curves.

**Figure 10 foods-12-01775-f010:**
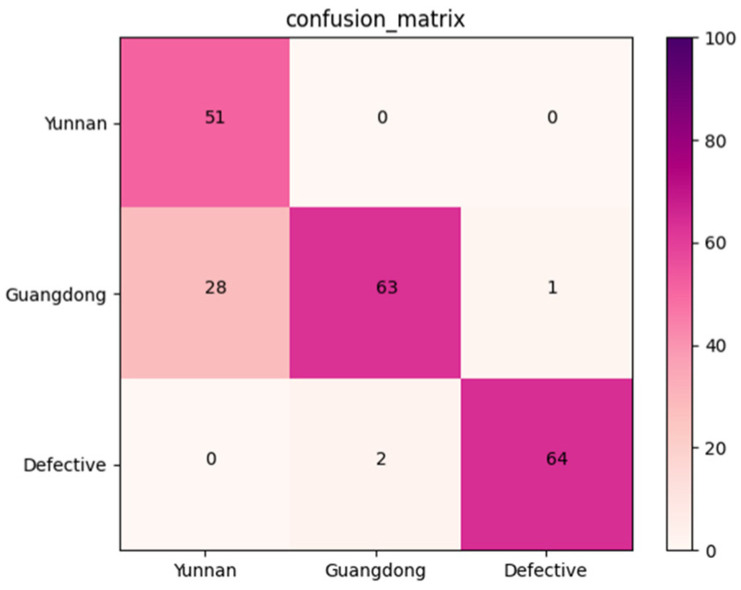
Confusion matrix of the proposed AFNet model in muti-category classification.

**Table 1 foods-12-01775-t001:** Composition and division of the dataset.

Dataset Name	Categories	Total Size	Proportion
quality classification	Normal	1430	3:1:1
	Defective	170	
origin identification	Yunnan	455	7:2:2
	Guangdong	594	
three-category classification	Yunnan	452	4:1:1
	Guangdong	587	
	Defective	101	

**Table 2 foods-12-01775-t002:** Architecture of the AFNet.

Layers	Number of Filters	Size of Filters	Stride
Input	-	-	-
1st Conv + Relu	4	3 × 3	1
Dropout	30%	-	-
2nd Conv + Relu	8	3 × 3	1
Dropout	40%	-	-
3rd Conv + Relu	16	3 × 3	1
MaxPooling	-	2 × 2	2
Dropout	50%	-	-
Flatten Layer	-	-	-
1st Dense Layer	-	-	-
Dropout	20%	-	-
2nd Dense Layer	-	-	-
Dropout	30%	-	-
3rdt Dense Layer	-	-	-
Dropout	30%	-	-
Output Layer + Softmax	-	-	-

**Table 3 foods-12-01775-t003:** The validation result of different CNN models.

No.	Model	Accuracy	Precision	Specificity
1	AFNet model	96.33%	96.27%	100.0%
2	BSSNet model	94.05%	95.56%	96.16%
3	VGG16 model	96.13%	96.86%	97.89%
4	Resnet18 model	94.33%	93.38%	99.29%
5	Inception model	95.87%	95.27%	99.29%

## Data Availability

The data presented in this study are available on request from the corresponding author.

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
