# Peer review of "Deep Learning Network of Amomum villosum Quality Classification and Origin Identification Based on X-ray Technology"

_foods, 2023, doi:10.3390/foods12091775_

Round 1

Reviewer 1 Report

The study focused on developing a non-invasive method to identify faulty Amomum villosum fruits using a combination of X-ray imaging technology and deep learning. To enhance the precision of evaluating the quality of Amomum villosum, the researchers devised a convolution neural network. Additionally, a novel algorithm was formulated to determine the source of Amomum villosum, surpassing the accuracy of the conventional convolutional neural network.

The idea is good however there are several concerns that should be considered:

Table 1 is confusing. It is difficult understand how the authors split the data.

Authors should give detail information about why they divide data as quality classification, origin identification, three-category classification.

Why the authors used different portions (3:1:1, 7:2:2, 4:1:1) for validation? This is not scientific! It sounds so weird!

What was the size of the images?

Criteria used for evaluating of performance of networks suit for binary class but is not for multi classes. Please look at the article: Sokolova, M., Lapalme, G. A systematic analysis of performance measures for classification tasks. Information processing & management, 2009; 45: 427-437. doi: 10.1016/j.ipm.2009.03.002

I don’t remember if BSSNet can be implemented in .Net. How did you manage it? Could you give detail information about it?

How did you implement AFnet that you build in .Net?

I highly recommend sharing the codes that you used for public use. You can considered it as supplementary material or you can upload it in the platform like GitHub. This is very important for increasing potential to get citation and impact of the journal.

These issues should be carefully addressed.

Reviewer 2 Report

The authors utilize a deep learning network to identify the origin of Amomum villosum and its quality based on X-ray technology. This research is quite interesting because of the way how the authors classify the quality of Amomum villosum and its origin using a deep-learning network.

The authors must prove their arguments on lines #34 to 35 with references.

The authors mentioned the usage of a deep-learning network for the safe use of Amomum villosum, but the following sentences do not support their claim in lines #39 to 40.

It would be great if the authors add references to support their argument in lines #48 to 49.

The authors need to reorganize the sentences in lines #60 to 63.

Figure 2 should be higher than the current resolution to identify letters clearly.

The authors need to check the policy of the journal on Keywords.

Reviewer 3 Report

I have read the Manuscript "Deep learning network of Amomum villosum quality classifica-2 tion and origin identification based on X-ray Technology” by Zhouyou Wu et al. The work considers a machine vision system based on a convolutional neural network (CNN) to sort Amomum villosum by using X-ray imaging technology (a non-destructive testing approach). The main aim is the identification of the internal structure for quality classification and origin identification.

The results achieved by implementing a new algorithm called “Amomum villosum fruit network” (AFNet) showed a very high accuracy implying the possibility to be used within innovative applications for improving industrial production efficiency.

The work is interesting, the results are sound and the conclusions are well supported by the achieved information. Furthermore, it is well written and also well organized. However, I noted that the reference section should be updated:

1)      also in consideration of the high importance of the topic from both the point of view of the approach and of the food industry, only 22 references are really few. Please include relevant references (also older for the development of CNN) and also the corresponding discussion. I.e.,

a)      10.5591/978-1-57735-516-8/IJCAI11-210

b)      10.23919/MIPRO48935.2020.9245376

c)      https://doi.org/10.3390/app112110301

2)      the format is not homogenous and sometimes erroneous/incomplete

Round 2

Reviewer 1 Report

It can be publishable. 

Reviewer 2 Report

The authors revised all my comments.